# Compression Strength of PLA Bolts Produced via FDM

**DOI:** 10.3390/ma15248740

**Published:** 2022-12-07

**Authors:** Mateusz Kukla, Igor Sieracki, Wojciech Maliga, Jan Górecki

**Affiliations:** Faculty of Mechanical Engineering, Institute of Machine Design, Poznan University of Technology, 60-965 Poznan, Poland

**Keywords:** polylactic acid (PLA), fused deposition modelling (FDM), nut and bolt, thread connection, storage modulus, loss modulus, compression modulus

## Abstract

The aim of this research was to define the compression strength of polylactic acid bolts produced using the fused deposition modelling method. In accomplishing this, static and cyclic compression tests for different metric thread sizes were carried out in accordance with ISO 4014. Tests were conducted on M42, M48, M56, M60, and M64 threads, while samples with three different types of pitch—one nominal and two fine threads—were prepared for each diameter. Standard ISO 604 for defining the compression modulus $E_{c}$ was implemented as the test basis. Accordingly, the mean compression modulus value $E_{c}$ for all measurements was 917.79 ± 184.99 MPa. Cyclic compression tests were then carried out on samples with the M64 × 4 thread. Fifty thread loading cycles were carried out for each variant to obtained different strain amplitude values and strain frequencies. Our work indicated that the values of the storage modulus defined in cyclic tests $E^{\prime }$ increased, while the values of the loss modulus $E^{\prime \prime }$ decreased when the value of the strain frequency increased. We found it not possible to determine the nature of the changes in the value of the storage modulus $E^{\prime }$ in the function of the strain amplitude. We did, however, observe an increase in the value of the loss modulus $E^{\prime \prime }$, together with the increase in the tested range of the strain amplitude. The determined mechanical values can be therefore be used for designing threaded connections made of polylactic acid using the fused deposition modelling method.

## 1. Introduction

Threaded connections represent one of the main ways of connecting elements in mechanical engineering. They are usually made of metallic materials and their alloys. This depends mainly on their application in the context of their strength, as well as on other material properties such as resistance to corrosion or to the exposure to acidic or alkaline environment. Threads are typically formed using machining or plastic-forming processes. However, progress in the field of material engineering and production techniques has created new possibilities for the manufacturing of threaded elements.

The progress in 3D printing technology has enabled the production of joining elements such as bolts and nuts and their practical application [1]: 3D-printed bolts are already in use in biomedical and industrial engineering, as well as in the production of toys, for example [2,3,4]. Due to the advantages of 3D printing, there has been an increase in the number of elements manufactured using this technology in recent years. In this context, conducting research on 3D-printed bolts and bolted connections seems logical.

Some research projects on 3D-printed bolts have been performed in the past, but in small numbers. In [2], Ramesh C. et al. analysed the dimensional precision of M12 × 1.5 bolts and their nuts made of VisiJet M2R-WT material. They also tested the shear strength of manufactured nuts and bolts. They observed dimensional deviation of the bolts and nuts, which depended on their positioning on the base plate during 3D printing. Nevertheless, the measured deviations were within the tolerance grade of ISO 2768 standards related to the General Tolerances Grade. They concluded that any orientation of the printed threaded elements in the lateral direction of the base plate can be used for mass production. However, strength tests conducted in [2] demonstrated that—depending on the positioning of the produced bolts and nuts on the base plate—variations in the density and shear strength are observable. Their positioning also influenced the mechanism of destruction of the elements (ductile or brittle failure).

The subject of the dimensional deviation of threaded elements produced using 3D printing techniques was also discussed by Ranjan N. [5]. He analysed M6 × 1 bolts with nuts made of ABS. Those tests demonstrated that the produced elements are compliant with the requirements of standards ISO 4014/4017. Moreover, the analysis of surface roughness values revealed values of less than 6 µm, which according to the author makes them suitable for commercial manufacturing.

In [6], Harshitha V. and Rao S.S. performed tests on bolts made of ABS and PLA materials (M12 × 1.75 bolts of dimensions compliant with standard ISO:421). They demonstrated that in the context of both sheer stress and equivalent stress, PLA bolts are stronger than those made of ABS. Similarly, Kumar C.L. et al. [7] performed tests for failure of bolts M12 × 1.75 made of ABS and PLA under shear stress conditions. On the basis of the obtained results, the authors concluded that bolts made of PLA are stronger and have a shear stress safety factor of 1.02, as opposed to 1.27 for ABS [7].

Wi J.H. et al. analysed the effect of self-loosening of 3D-printed bolts, using M12 bolts of the length of 70 mm [1]. They compared a bolt made of Procisa 779 resin and manufactured using the stereolithography method with a steel bolt. The results indicated that the steel bolt is more durable during the first stage of loosening. In addition, connectors produced using the 3D printing technology exhibited a clamping force of less than 30%. The authors of this research concluded that this is most probably due to the very nature of the used material. They also indicated that a 3D-printed bolt can be more effective in a threaded connection in which a special shape is more essential [1]. Eraliev O. et al. analysed the self-loosening phenomenon of 3D-printed bolts under cyclical changes of temperature during operation [8]. The experiments were performed using M12 × 1.75 bolts fabricated in accordance with the ISO standard and made of acrylonitrile butadiene styrene (ABS-2), polylactic acid (PLA) and glass. The results of these tests demonstrated that the glass bolts exhibited the lowest performance in high temperature changes. Moreover, bolts made of PLA revealed poor performance in the aspect of self-loosening in low temperature changes, while the ABS-2 bolts showed good performance under cyclical temperature changes. However, bolts made of this material were weaker than the other ones in the context of maximum preload [8].

Threaded 3D-printed elements seem to be especially advantageous in non-standard applications. This is especially true when it is necessary to obtain a large or an unusually shaped threaded element intended to withstand low or medium loads. In view of the above, the aim of this research was to determine the compression strength of PLA bolts produced using the FDM method. For this purpose, static and cyclic compression tests were carried out on metric threads of selected sizes. In view of the above, the aim of this research was to determine the compression strength of PLA-made bolts produced using the FDM method. For this purpose, static and cyclic compression tests were carried out on metric threads of selected sizes. It was considered crucial to determine the values of compression modulus $E_{c}$, storage modulus $E^{\prime }$ and loss modulus $E^{\prime \prime }$. Threaded connections usually experience axial force with a static or variable value resulting in varying strain amplitude and frequency of load. It is therefore reasonable to analyse material parameters in this aspect. Previous literature studies, in terms of axial stress, focus mainly on the tensile strength of PLA. For this reason, the analysis of compressive strength is justified, which complements the state of knowledge in this field.

Current applications of PLA bolts and their composites include medical aspects such as PLA bone screws containing iron oxide (Fe3O4) nanoparticles [9], PLA in fixation plates and screws [10], pins, suture anchors, and screws composed of PLA and copolymers [11]. One can also indicate applications in toys and toy elements [8]. It is possible that further studies of this type of connection will contribute to their wider use in future.

The choice of PLA can be justified because in the case of fused deposition modelling (FDM), it is one of the most commonly used materials. It has many advantages, including biodegradability, biocompatibility and environmentally friendly properties [12,13]. At the same time, the use of pure PLA also has disadvantages: among others, relatively low mechanical properties, water-solubility rate, and high brittleness, as well as low heat resistance [12,13].

## 2. Materials and Methods

### 2.1. Sample Preparation

For the purpose of the tests, the authors prepared samples in the form of threaded rods in a shape that was compliant with standard ISO 4014 [14]. They were printed via FDM technology, using polylactic acid (PLA) material. Filament Fiberlogy Refill Easy PLA (Fiberlab S.A., Brzezie, Poland) was employed to produce the test samples. This is a commercially available material that can be considered ‘typical’. Mechanical properties of this material declared by the supplier are presented in Table 1.

In order to determine the crystallographic properties of the samples, X-ray diffraction (XRD) was carried out. This was conducted using a powder diffractometer (SmartLab Rigaku, Tokyo, Japan) with a CuK α lamp, in the angle range of 3–80° (2θ), in steps of 0.04°. A square print sample with a side length of 20 mm was selected for testing. This sample was made with the same printing parameters as the tested elements. However, it was manufactured as a solid element (100% infill rate) for test accuracy and in order to achieve the greatest certainty of results as possible. The XRD measurements were performed at room temperature. The test results are shown in Figure 1. As can be seen in the diffraction pattern, the PLA filament used in 3D printing is not completely amorphous. Amorphousness is indicated by the presence of a broad diffraction peak at the maximum at the angle of 16.1° (2θ) and at the angle of 33.3° (2θ) [15]. Reflections present at 27.2° angles; 35.5°; 55.0° (2θ) may indicate the presence of coloured PLA additives [16].

Size M42, M48, M56, M60 and M64 threads were selected for printing. For each of these diameters, the authors prepared samples with three different pitch types—one nominal and two fine threads. The length of the threaded rod was adapted to equal the triple height of its corresponding nut (Figure 2a; according to ISO 4032 [17]). This was dictated by the need to enable two nuts to be entirely screwed onto the ends of the threaded rod. As a result, the free length of the thread (unoccupied by nuts) was approximately equal to the height of one nut. The parameters of the prepared samples are shown in Table 2.

After the threaded rods were printed, they were measured in order to verify their major and minor diameters, as well as pitch. Each of the prepared samples was examined in randomly selected points. All diameters were confirmed to be with the range defined by the nominal dimension ±0.5 mm. An additional measurement of the distance of ridge grooves between 5 and 10 ridges (also in randomly selected points) was made, the results of which were then added and divided by the number of the measured ridges. All of the pitches assessed as above were confirmed to be within the range defined by the nominal dimension ±0.1 mm. A 3D model of the threaded rod was prepared using Autodesk Inventor 2019 software (Autodesk Inc., One Market Plaza, San Francisco, CA, USA) and the threadModeler add-on. Translation of the 3D model into individual layers (slicing) was accomplished with Ultimaker Cura 4.8.0 software. The parameters applied during printing are shown in Table 3. In order to achieve sample effective weight-to-strength ratio, 35% filling with triangle pattern was applied [18,19]. The implemented layer height was selected experimentally. The printing of several test samples allowed the authors to select a layer height that prevented the occurrence of layer separation.

**Table 1 materials-15-08740-t001:** Properties of tested materials (based on [20]).

Property	Testing Method	Value
Specific Density	ISO 1183	1.24 g/cm^3^
Tensile Strength (Yield)	ISO 527	50 MPa
Tensile Strength (Break)	ISO 527	53 MPa
Tensile Modulus	ISO 527	3500 MPa
Elongation (Yield)	ISO 527	6%
Flexural Strength	ISO 178	81 MPa
Flexural Modulus	ISO 178	3800 MPa
Izod Impact Strength (Notched, 23 °C)	ISO 180	2 kJ/m^2^

**Table 2 materials-15-08740-t002:** Selected parameters of the thread and nuts.

Diameter d (mm)	Pitch P (mm)	Pitch Diameter $d_{2}$ (mm)	Minor Diameter $d_{3}$ (mm)	Infill Diameter $d_{4}$ (mm)	Nut Height h (mm)	Diameter of the Hole in the Nut $D_{1}$ (mm)
42	1.5	41.026	36.479	34.679	34	40.376
	3	40.051	38.319	36.519	34	38.752
	4.5	39.077	39.479	37.679	34	37.129
48	2	46.701	45.546	43.746	38	45.835
	3	46.051	44.319	42.519	38	44.752
	5	44.752	41.866	40.066	38	42.587
56	2	54.701	53.546	51.746	45	53.835
	4	53.402	51.093	49.293	45	51.67
	5.5	52.428	49.252	47.452	45	50.046
60	2	58.701	57.546	55.746	48	57.835
	4 *	57.402 *	55.093 *	53.293 *	48 *	55.67 *
	5.5	56.428	53.252	51.452	48	54.046
64	2	62.701	61.546	59.746	51	61.835
	4	61.402	59.093	57.293	51	59.67
	6	60.103	56.639	54.839	51	57.505

* Samples of this size were also used for cyclical tests.

**Table 3 materials-15-08740-t003:** Printing parameters.

Parameter	Value	Parameter	Value
Layer height	0.15 mm	Fill printing speed	70 mm/s
First layer height	0.2 mm	Exterior wall printing speed	45 mm/s
Layer width	0.45 mm	Outer layers printing speed	45 mm/s
First layer width	0.45 mm	Travel speed without printing	80 mm/s
Number of layers in the wall	4	Top and bottom layer printing speed	32.5 mm/s
Distance between lines	0.2 mm	Number of slower layers	2
Bottom and top thickness	0.8 mm	Acceleration of printing	1500 mm/s^2^
Number of top layers	3	Acceleration of travel speed without printing	2000 mm/s^2^
Number of bottom layers	5	Retraction	Yes
Infill ratio of the rod core	35%	Retraction distance	3 mm
Filling shape	Tringle pattern	Retraction speed	25 mm/s
Printing temperature	208 °C	Retraction acceleration	25 mm/s^2^
First layer printing temperature	208 °C	Retraction for minimal movement	0.9 mm
Initial printing temperature	198 °C	Applicable for	All layers
Worktable temperature	60 °C	Cooling fan efficiency grade	100%
Printing speed	65 mm/s	Initial height without maximum cooling	5 mm

### 2.2. Methodology of Static Tests

The tests were based on standard ISO 604 [20], which defines the mechanical properties of plastics during compression. Certain changes were made in the experimental setup due to the use of threaded rods for testing. The tests were undertaken on an MTS Insight testing machine (MTS Systems Corporation, Eden Prairie, MN, USA) that was equipped with a 50 kN measurement load cell. The tests were carried out in laboratory conditions at a temperature of 25 °C and humidity of 40%. Five samples of each selected size were assessed (see Table 2). The samples were compressed at a rate of v = 3.5 mm/min until their destruction or the achievement of the load of 47 kN.

A diagram and view of the testing station is shown in Figure 2a. The testing machine (1) has a clamping jaw (7) that contains an aligning element (6) made of two cooperating spherical caps, which have a large diameter of curvature and very low surface roughness. They enable the self-centring of the compressed configuration in order to ensure that the force acting on the tested sample is as close to an axial force as possible. The aligning element (6) holds cylinder (3), which supports the threaded rod configuration (5), together with the nuts (4) that are screwed onto the rod. The upper nut (4) supports another cylinder (3), which is compressed by the load cell (2) that enables the measurement of the generated force F. The machine simultaneously measures the displacement x of the load cell (2) in relation to its initial position. Figure 2b presents a diagram of the tested configuration.

Compressive stress σ and strain ε values were determined on the basis of the measured compressive force F and displacement x values. The effect of the deformation of steel elements in the tested configuration was ignored in the calculations, as their rigidity is several orders of magnitude higher than that of the tested material. Therefore, the error generated by these elements can be considered marginal.

The cross-section of the threaded rod is defined by the following geometrical values: pitch diameter $d_{2}$, minor diameter $d_{3}$ and infill diameter $d_{4}$. Because the infill ratio of the rod core was 35%, the core surface area of the threaded rod is defined by Equation (1):(1)$$A=\frac{\pi }{4}\left[d_{3}^{2}-0.65d_{4}^{2}\right]$$

On this basis, the authors determined compression stress–strain curves and then defined the material parameters in accordance with the procedure of standard ISO 604 [21]. The values of compression modulus $E_{c}$ were determined using Equation (2):(2)$$E_{c}=\frac{\sigma_{2}-\sigma_{1}}{\varepsilon_{2}-\varepsilon_{1}},$$
where $\sigma_{1}$ is the stress value measured for strain $\varepsilon_{1}$ = 0.0005, whereas $\sigma_{2}$ is the stress value measured for strain $\varepsilon_{2}$ = 0.0025.

The authors also determined the value of compression force $F_{3\%}$ registered during the experiment, which corresponded to strain value of $\varepsilon_{3\%}$ = 0.03. This value was determined experimentally, as it was the point beyond which the compression stress and strain ratio were not in the proportional range (see Figure 3). Values of $F_{3\%}$ were used to calculate the values of the corresponding stress $\sigma_{3\%}$, force applied to one active thread ridge cooperating with the nut $F_{n}$ and the value of contact stress (pressure) p that was present at the interface between the lateral surfaces of the steel nut and the PLA threaded rod. The indicated values were calculated according to Equations (3) and (4):(3)$$F_{n}=\frac{h}{p}\cdot F_{3\%},$$
(4)$$p=\frac{4p\cdot F_{3\%}}{\pi hz\cdot \left[d^{2}-D_{1}^{2}\right]},$$
where z is the number of starts of thread (single-start threadform for metric screw). The arithmetic average of five measurements was implemented as the estimator of calculated values $E_{c}$ and $\sigma_{3\%}$. Standard deviation of the arithmetic average was implemented as the error of the calculated value.

### 2.3. Methodology of Cyclic Tests

The second type of performed tests involved the cyclic loading of samples. The methodology of these tests was similar to the methodology of static tests, apart from the exceptions described in this chapter. Threaded rod samples of the dimensions M60 × 4 (see Table 2) were cyclically compressed in the course of n = 50 cycles. Compression tests were carried out for five different strain frequencies (between $f_{1}$ and $f_{5}$) and four different strain amplitudes (between $\varepsilon_{01}$ and $\varepsilon_{04}$) and the individual values are shown in Table 4.

Before testing, the initial load $\sigma_{i}$ of 0.45 MPa was applied to the samples. This value was determined on the basis of Equation (5), specified in standard [21]:(5)$$0\leq \sigma_{i}\leq 5\cdot 10^{4}\cdot E_{c},$$

The average value of the compression modulus $E_{c}$ was assessed during static tests and will be described in the following chapters.

This way, series of the values of compressive force F and displacement x as a function of time were collected for 50 cycles. These values were then used to determine the functions of compressive stress σ and strain ε, using the same methodology as previously on the basis of Equation (1). Average values for these cycles were then calculated by adding together 5 successive value series. Hence, 10 average values for cyclic changes in compressive stress σ and strain ε values over time were obtained, which can be expressed as a series of sums defined by Equation (6):(6)$${\bar{x}}_{1}=\frac{1}{5}\overset{5}{\underset{i=1}{\sum }}n_{i}\left(t\right),{\bar{x}}_{2}=\frac{1}{5}\overset{10}{\underset{i=6}{\sum }}n_{i}\left(t\right),\ldots$$
where ${\bar{x}}_{1}-{\bar{x}}_{10}$ are the average values for successive series, $n_{i}$ are successive stress–strain values, whereas i is the successive cycle number. This methodology was implemented to accelerate further calculations and to simplify the analysis of data. Due to the scale of the observed changes, it seems that the comparative analysis of individual cycles would be meaningless.

The obtained series of values were used to define the parameters of a viscoelastic model material (the Kelvin–Voigt model). For this purpose, the values of loss angle φ were defined on the basis of the characteristic points of the stress–strain mechanical hysteresis loop. The determination of this value allowed the authors to calculate the values of the storage modulus $E^{\prime }$ and loss modulus $E^{\prime \prime }$ on the basis of Equation (7):(7)$$\frac{\sigma_{0}}{\varepsilon_{0}}cos\varphi =E^{\prime }\;\mathrm{and}\;\frac{\sigma_{0}}{\varepsilon_{0}}sin\varphi =E^{\prime \prime }.$$
where $\sigma_{0}$ is the stress amplitude, $\varepsilon_{0}$ is the strain amplitude.

## 3. Results and Discussion

### 3.1. Results of Static Tests

Figure 3 demonstrates examples of test results for selected (for the sake of clarity) samples of the dimensions M42 × 1.5. Figure 4 presents the defined values of compression modulus $E_{c}$ divided into specific diameters and thread pitches. The lowest registered value was 673.08 ± 23.77 MPa for size M64 × 6, whereas the highest value was 1257.51 ± 34.32 MPa registered for size M64 × 6 MPa, which represents a difference of almost 87%. The mean compression modulus value $E_{c}$ for all measurements was 917.79 ± 184.99 MPa. An analysis of these results reveals that they exhibit a rather large spread of values.

There are a number of research papers that deal with the assessment of the value of compression modulus $E_{c}$ for samples made of PLA. The authors of [22] determined the value of this coefficient at approximately 1750 MPa, without registering any significant effect connected with flat and upright printing orientations (cylindrical sample of the diameter of 12.7 mm and height of 25.4 mm, with the compression velocity of 1.3 mm/min). Research described in [23], on the other hand, demonstrated compression modulus $E_{c}$ values of approximately 2578 MPa for out-of-plane compression tests and approximately 2029 MPa for in-plane compression tests (rectangular samples, base side length 12.7 mm, height 40 mm). These were similar to values determined in [24]. The authors of this paper tested rectangular samples with base dimensions 10 × 4 mm and the height of 50 mm, with the compression velocity of 6 mm/min. The average value of compression modulus $E_{c}$ determined in these tests was approx. 2529 MPa. Work described in [25] included the testing of rectangular samples with a base side length of 12.7 mm and heights between 20 mm and 65 mm, with the compression velocity of 1.3 mm/min. The researchers observed that the value of compression modulus $E_{c}$ increases proportionally to the length of the sample, with the exception of several local oscillations. Furthermore, the lowest recorded value was 1951 MPa, whereas the highest was 2313 MPa. The influence of the height of the tested samples on the values of $E_{c}$, indicated in [25], cannot be clearly confirmed on the basis of the collected data, as the heights of threaded rods made of PLA were significantly different in terms of the longest and shortest value (the rod’s height was three times the height of the nut). The length of the measurement section, however, always equalled the height of the nut, and thus the differences in the lengths of the measurement sections themselves were smaller than between the lengths of entire rods. Due to the high rigidity of the nut, it is assumed that the highest strain occurred at this very section of the sample.

We can therefore conclude that the results were also significantly affected by factors other than the lengths of the sample measurement sections themselves. The infill of the rods was always generated in the same way and with the same orientation within the printing area, so the observed differences probably cannot be attributed to the printing method. Perhaps, however, they are generated by the quality of the mutual bonding of the individual 3D print layers. This issue remains open and requires further testing.

An analysis of the presented values has revealed that the registered results differ from the data presented in the literature by 2–2.5 times. This is most probably due to the implemented infill ratio of 35%. The most similar test results that can be found are those in [26]. The authors of these tests analysed cubic samples with a side length of 25 mm, wall thickness of 0.8 mm and compression speed of 1 mm/min, which had various infill ratios and profiles. In the case of samples made of PLA with a triangle pattern and infill density of 40%, 60% and 80%, compression modulus $E_{c}$ values of approximately 567 MPa, 779 MPa and 1052 MPa were registered, respectively. Samples registered in [26] were compressed transversely to the triangular structure (in the course of tests presented in this article, compression was conducted longitudinally to the inner triangular structure—see Figure 2b). Considering this fact, as well as the differences in the thickness of the sample walls, these values can be considered somewhat similar. This seems to confirm the influence of the infill ratio of 35% on the differences in the registered compression modulus $E_{c}$ values in comparison to solid PLA samples.

Figure 5 shows stress values $\sigma_{3\%}$ that correspond to the strain value $\varepsilon_{3\%}$ = 0.03 and contact stress p that is present at the interface between the lateral surfaces of the steel thread of the nut and the threaded rod. These two values are mutually correlated through the value of force $F_{3\%}$, in accordance with Equations (3) and (4). The analysis of the graph for all obtained results demonstrates that values $\sigma_{3\%}$ and p decrease proportionally to the increase in the diameter for the individual thread pitches listed in ascending order. This is most probably caused by the increasing diameter of the cross section of successive samples. However, the observed changes are not linear in nature. The recorded values differ 2- to 2.5-fold from those that can be found, for example, in [23,24]. Once again, this is most probably caused by the effect of the selected fill ratio.

Figure 6 shows the calculated values of the force applied to one active thread ridge that cooperates with the nut $F_{n}$. Graph data analysis demonstrates that the values of force $F_{n}$ increase proportionally to the decrease in the value of pitch P, which results in the decreasing number of thread ridges along the height of a given nut h. The exception to this rule is a value recorded for thread M56 × 4.

Of all the tested threaded rods, eight suffered permanent damage (about 11% of all samples). Figure 7 illustrates a view of selected damaged threaded rods. One of the eight samples in Figure 7a exhibits a very characteristic type of failure (M60 × 4). One can observe the presence of fractures parallel to the direction of the applied compressive force. An analysis of the sample surface revealed sharp edges on the fracture surface. This kind of failure is most likely caused by shearing between layers because of inadequate interlayer bonding [2]. The remaining samples suffered ductile failure, which revealed itself in the form of the outflow of material. All eight threaded rods were damaged in the area of their middle sections—in an area that does not cooperate with the nuts and is therefore subject to the highest strain. Values defined in this part of the research are shown in Appendix A as Table A1.

### 3.2. Results of Cyclic Tests

Figure 8 illustrates how the maximum value of the registered force changes with the number of load cycles. The maximum value of the force decreases especially for the first five cycles, which is connected with the stabilisation of the material of the tested samples. The presented graph also reveals that increasing the velocity of compression changes the nature of the registered value trend. Herein, increasing the velocity generates higher oscillation of the maximum values, while for lower velocities, the trend is flatter.

Figure 9 and Figure 10 illustrate the changes in storage modulus $E^{\prime }$ and loss modulus $E^{\prime \prime }$ for the tested frequency and strain amplitude values. The list of obtained values is shown in Table A2.

An analysis of Figure 9 demonstrates that the values of the storage modulus $E^{\prime }$ increase, while the values of the loss modulus $E^{\prime \prime }$ decrease when the value of the strain frequency increases. This is typical of most solid polymers. This could be rationalized on the example of moving from viscous-like behaviour (there is greater time available for viscous flow at low frequencies) to solid-like behaviour (at frequencies of higher value) [27].

An analysis of Figure 10 did not explicitly determine the nature of the changes in the value of the storage modulus $E^{\prime }$ as a function of the strain amplitude. However, loss modulus $E^{\prime \prime }$ values in the analysed range increased together with the increase in strain amplitude values (in some cases, as much as double).

On the basis of data in available literature, it can be demonstrated that the values of the storage modulus $E^{\prime \prime }$ for melted PLA without any additives are not significantly affected by the strain amplitude [28]. Indeed, the values of the storage modulus $E^{\prime }$ PLA for very small strain amplitude values are more or less constant, but they decrease for high strain values [29] (in this case for shearing). This effect can change with the introduction of additional materials into the PLA material in order to obtain a composite. In such cases, one can observe the dependency of dynamic modules on the strain amplitude in the case of increasingly small values (depending on the composition of the composite). This is true, for example, for PLA–cellulose nanocrystals composites [29], PLA–poly(caprolactone) blends [30], PLA–polycaprolactone, PLA–poly(butylene adipate-*co*-terephthalate), and PLA–poly(butylene succinate-*co*-butylene adipate) [31].

However, in the discussed case, PLA without any additives was analysed. The observed changes were perhaps caused by the fact that the printed structure was not solid and represented a triangle pattern infill structure (see Figure 2b) and was therefore susceptible to strain. Still, one cannot rule out that the general nature of the observed changes is unknown due to the low values of the implemented strain amplitude. This issue remains open and requires further testing.

The authors were not able to find any results of other experiments that would be comparable with those presented in this article. A direct comparison was therefore not possible. Nevertheless, an attempt was made to refer to literature values.

In [32], the authors determined the storage modulus $E^{\prime }$ for PLA on the basis of a three-point bending test. The test was carried out for the frequency of f = 1 Hz and for the strain amplitude of $\varepsilon_{0}$ = 0.03. The samples had a rectangular shape and the dimensions of 30 mm × 12.7 mm × 1.8 mm to 2.5 mm, and the span was 25 mm. A storage modulus of approximately $E^{\prime }$ = 1050 MPa was recorded at 40 °C for samples that were thermally annealed and 1100 MPa for samples that did not undergo any thermal treatment (rotomoulded samples). In the case of samples produced by compression moulding, values of about $E^{\prime }$ = 1025 MPa were reported for treated and untreated samples. These values, despite being obtained in the course of a different test, demonstrate a similar range of values to those obtained during the presented tests.

In [33], the authors determined the properties of PLA using a dynamic mechanical analyser equipped with a dual-cantilever bending fixture. Samples of dimensions of 54.61 mm × 12.7 mm × 3.175 mm (2.15 in × 0.5 in × 0.125 in) were tested with a frequency of f = 1 Hz. At a temperature of 20 °C, values of the storage modulus of about $E^{\prime }$ = 3187 MPa and loss modulus of about $E^{\prime \prime }$ = 61 MPa were recorded (read from the graph).

Similarly, in paper [34], the authors used a dynamic analyser for the analysis of PLA. Experiments were performed in a multi-frequency strain modulus in a dual-cantilever clamp. For this purpose, the authors used frequencies of 1, 3, and 10 Hz and a strain amplitude of 10 µm. The samples were rectangular, with dimensions of 35 mm × 10 mm × 4 mm. At a temperature of 20 °C, storage modulus of about $E^{\prime }$ = 3169 MPa and loss modulus of about $E^{\prime \prime }$ = 33 MPa were recorded (read from the graph).

The analysis of the presented values has revealed that the registered storage modulus $E^{\prime }$ values differ from those presented in [32,33] by about 2.5–3 times. As in the case of static measurements, this is most probably caused by the implemented infill ratio. The discussed values are probably also affected by the implemented measurement methods. The values of the loss modulus $E^{\prime \prime }$, on the other hand, are in a similar range.

## 4. Conclusions

The conducted research and analyses allow us to conclude that:During the performed static compression tests, approximately 11% of all tested samples were damaged. Both fracture failure (1 case) and ductile failure (all other cases) of the samples was observed. Thus, even in uniform conditions of production of threaded elements using the FDM method, it is difficult to explicitly predict the nature of failure or the value of the breaking stress. It is thus advisable to consider the value of safety coefficient when implementing bolt connections made of PLA.The mean compression modulus value $E_{c}$ for all measurements was 917.79 ± 184.99 MPa. The large value of standard deviation further supports the conclusion of the difficulty of precisely predicting the strength parameters of a printed thread element.The defined values of the compression modulus $E_{c}$ differ from those specified in the literature by 2–2.5 times. The differences mentioned above were probably caused by the implementation of a threaded rod infill ratio of 35%.The values of the storage modulus defined in cyclic tests increase $E^{\prime }$, while the values of the loss modulus $E^{\prime \prime }$ decrease when the value of the strain frequency increases.Simultaneously, ascertaining the nature of the variations in storage modulus $E^{\prime }$ values in the function of the strain amplitude was not possible on the basis of the determined values. This issue remains open and requires further testing.

The importance of the results obtained is emphasized by the fact that they complement the knowledge in the field of axial loads of elements made by 3D printing from PLA. This particularly concerns threaded connections, which can constitute a promising prospect in certain applications. The determined mechanical values can be used for designing threaded connections made of polylactic acid using the fused deposition modelling method.

## Figures and Tables

**Figure 1 materials-15-08740-f001:**
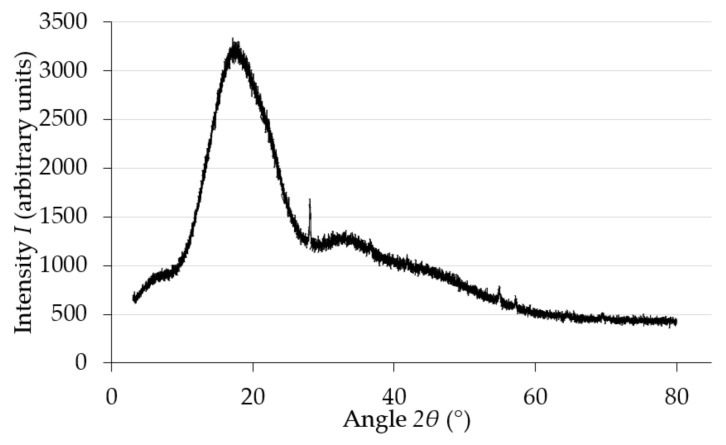
Diffractogram obtained for the tested PLA filament.

**Figure 2 materials-15-08740-f002:**
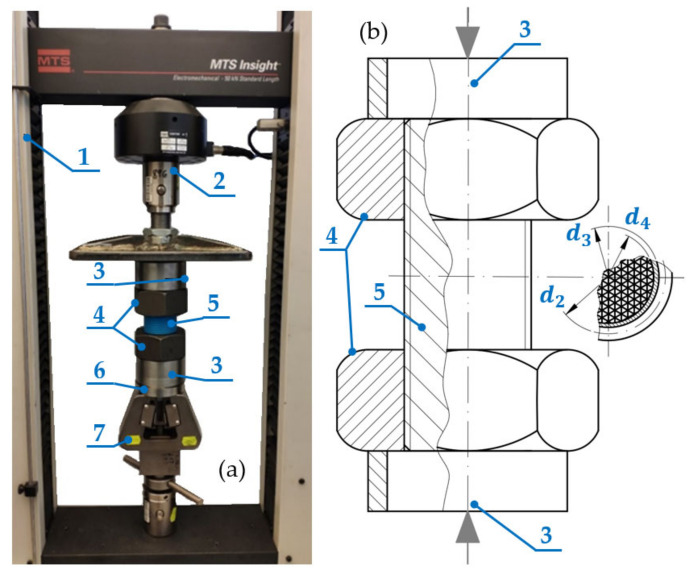
The test setup; (**a**) view, (**b**) scheme; 1—MTS testing machine, 2—load cell, 3—cylinder, 4—nut, 5—threaded rod, 6—aligning element, 7—clamping jaw.

**Figure 3 materials-15-08740-f003:**
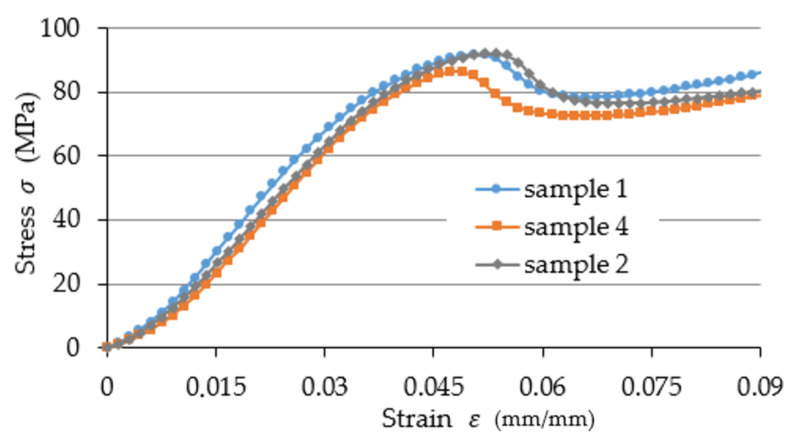
Examples of test results for selected (for the sake of clarity) samples of the dimensions M42 × 1.5.

**Figure 4 materials-15-08740-f004:**
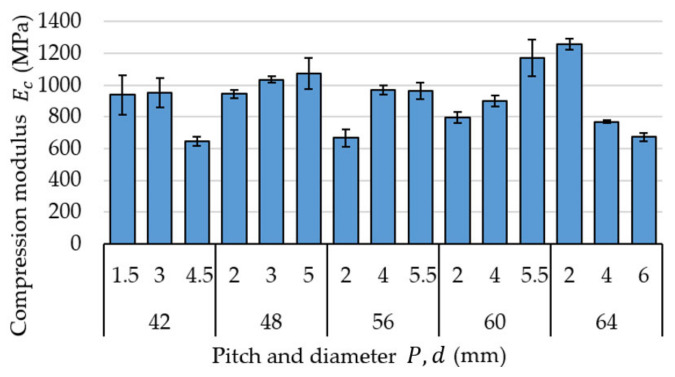
Defined values of compression modulus $E_{c}$ divided into specific diameters and thread pitches.

**Figure 5 materials-15-08740-f005:**
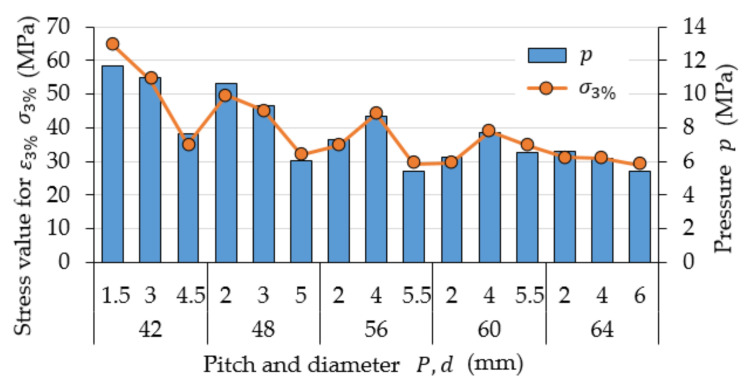
Stress values $\sigma_{3\%}$ that correspond to the strain value of $\varepsilon_{3\%}$ = 0.03 and contact stress p that is present at the interface between the lateral surfaces of the steel thread of the nut and the threaded rod.

**Figure 6 materials-15-08740-f006:**
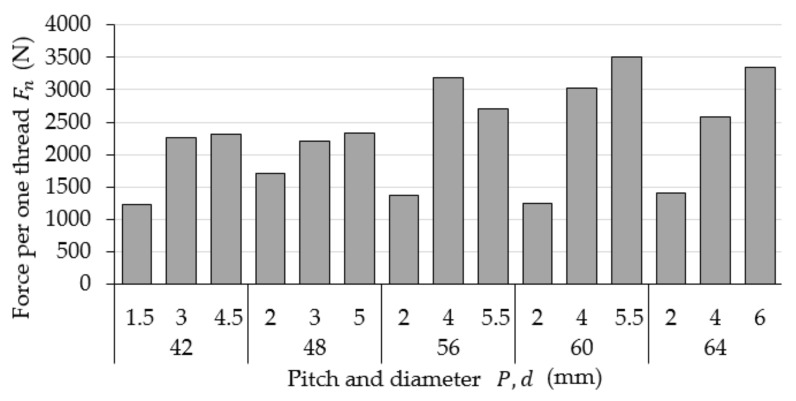
Calculated values of the force applied to one active thread ridge that cooperates with the nut $F_{n}$.

**Figure 7 materials-15-08740-f007:**
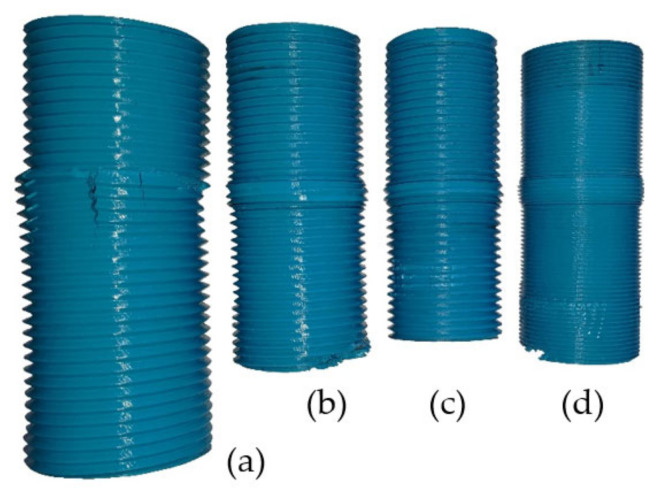
The view of selected damaged threaded rods; (**a**) fracture failure, (**b**–**d**) ductile failure; damage in the initial and final sections of rods (**b**) and (**d**) were made after the tests to assess the quality of the internal structure (they are not the result of the experiment).

**Figure 8 materials-15-08740-f008:**
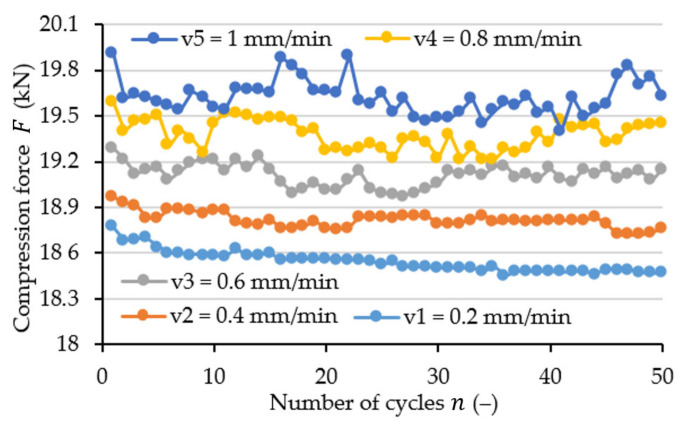
Change in the maximum value of the recorded force for different compression velocities and strain amplitude of $\varepsilon_{01}$ = 0.01 (mm/mm).

**Figure 9 materials-15-08740-f009:**
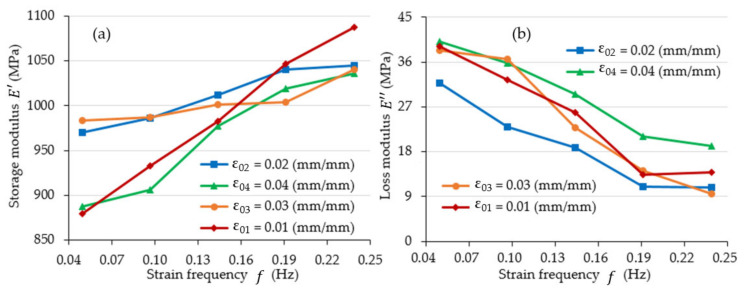
The value of: (**a**) storage modulus $E^{\prime }$ and (**b**) loss modulus $E^{\prime \prime }$ as a function of strain frequency f.

**Figure 10 materials-15-08740-f010:**
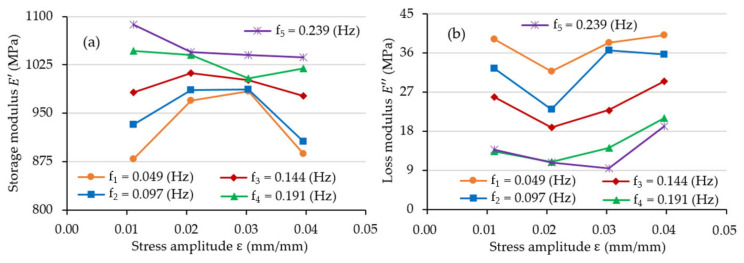
The value of (**a**) storage modulus $E^{\prime }$ and (**b**) loss modulus $E^{\prime \prime }$ as a function of strain amplitude ε.

**Table 4 materials-15-08740-t004:** Cyclic test parameters.

Strain Amplitude	Value (mm/mm)	Strain Frequency	Value (Hz)
$\varepsilon_{01}$	0.01	$f_{1}$	0.049
$\varepsilon_{02}$	0.02	$f_{2}$	0.097
$\varepsilon_{03}$	0.03	$f_{3}$	0.144
$\varepsilon_{04}$	0.04	$f_{4}$	0.191
		$f_{5}$	0.239

## Data Availability

The data that support the findings of this study are available from the corresponding author upon reasonable request.

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
