# Peer review of "Compression Strength of PLA Bolts Produced via FDM"

_materials, 2022, doi:10.3390/ma15248740_

Round 1

Reviewer 1 Report

1. Title can be modified to Compression strength of PLA made bolts produced by FDM method.

2. What is the novelty in this work? Please explain.

3. What is the use of producing bolts by FDM? The technique is highly costly.

4. Where are the PLA bolts used?

5. What is the basis for choosing strain amplitude and frequency. Please explain the scientific method behind choosing the parameters.

6. Figure 2. Why only samples 1,4,2 are reported. Why not others? What are samples 1,2,4?

7. What does Stress values σ_(3%) signify?

8. Figure 5. Calculated values of the force applied to one active thread ridge that cooperates with the nut F_n. Use subscripts wherever required. 

9. Storage modulus of about ?′ = 3221 MPa. Please give reference papers wherein such high values are reported. I really doubt there is an error in the values reported. 

10. Conclusions are exhaustive. The need to be crisp and precise. Please rewrite the conclusions in point form.

11. A thorough English language and grammar checking is required for the entire manuscript.

12. Use of subscript and superscript is to be corrected throughout the manuscript and in the figures. 

13. Figure 9. The value of (a) storage modulus E^' and (b) loss modulus E^'' as a function of strain amplitude ε. Please check the title of the fig.9.

14. The trend in fig. 9 is erratic. What is the reason for this? 

Author Response

The Authors appreciate the time and effort and are grateful for the insightful comments and valuable improvements to our work. We carefully considered the comments and tried our best to address all of them. In addition, many other corrections have been made. All changes are marked in “change tracking”. Detailed responses for the comments:

  1. Title can be modified to Compression strength of PLA made bolts produced by FDM method.

Corrections were added as suggested.

  1. What is the novelty in this work? Please explain.

The novelty of this work concerns experimental research on threaded connections made by 3D printing from polymeric materials - as opposed to the classical approach (plastic processing or machining from metal materials). Due to the advantages of 3D printing, there has been an increase in the number of elements manufactured using this technology in recent years. In this context, conducting research on 3D-printed bolts and bolted connections seems logical. The authors were unable to find a substantial amount of research related to this topic. The ones that were found were described in the introduction. Research concerning, in principle, dimensional accuracy and a small degree of mechanical properties can be found in [5-7]. Also in [1] self-loosing of 3D printed bolts was analyzed. The presented results are therefore a unique development of the topic of fasteners made using the 3D printing method - one could guess, this was one of the reasons for the problems with comparative analysis to other studies.

[1]. Wi, J-H.; Lee, K-H.; Lee, C-H. Self-loosening of 3D printed bolted joints for engineering applications, MATEC Web of Conf. 2018, 185, 00029, 1–6. DOI: https://doi.org/10.1051/matecconf/201818500029

[5]. Chand, R.; Sharma, V.S.; Trehan, R.; Gupta, M.K. A physical investigation of dimensional and mechanical characteristics of 3D printed nut and bolt for industrial applications, Rapid Prototyping J., 2021, 28(5), 953–966(14). DOI: https://doi.org/10.1108/RPJ-09-2021-0250

[6]. Ranjan, N. Dimensional and roughness analysis of ABS polymer-based 3D printed nuts and bolts.            Mater. Today: Proc. 2022, 48(5), 1604–1608. DOI: https://doi.org/10.1016/j.matpr.2021.09.492

[7]. Harshitha, V.; Rao, S.S. Design and analysis of ISO standard bolt and nut in FDM 3D printer using PLA and ABS materials. Mater. Today: Proc. 2019, 19(3), 583–588. DOI: 10.1016/j.matpr.2019.07.737

  1. What is the use of producing bolts by FDM? The technique is highly costly.

Indeed, from the point of view of industrial mass production, producing screws using the FDM technique is economically unprofitable. However 3D-printed threaded elements seem to be especially advantageous in non-standard applications. This is especially true when it is necessary to obtain a large or an unusually shaped threaded element, intended to withstand low or medium loads (for example, for the production of prototypes or ornamental elements). What's more, dedicated unit production using FDM methods in combination with the properties of PLA (biodegradability, biocompatibility) makes this type of connection a promising prospect in medical applications.

In this matter from [2], a direct quote could be recommended:

The accuracy and quality of the manufactured nut and bolt are directly and indirectly related to all safety problems and self-loosening. Nut and bolt quality and precision because of recent advances in 3D printing technology, it is now possible to fabricate complex components on-demand. Fasteners, like 3D printed bolts, may be used in the fabrication of 3D printed objects. These may be very beneficial. The Medical Industry and Medical Education have also been transformed by additive manufacturing. Since now, composite-based bolts have been used in place of titanium screws, which were previously used. It has also been discovered that 3d printed Screws perform better than titanium bolts in terms of efficiency and strength for knee surgical implants. Additively manufactured organs are used to educate medical students for surgical procedures, but they are also used in research. Because the organs are comparable to human body parts in terms of form, size, appearance, emotions, etc., additively manufactured tissues are on the way, and they will keep astronauts healthy throughout space trips. Several major space organizations have begun work experiments in this area (Muruga et al., 2019; Wi et al., 2018). For polymer-based 3D printing, multi-jet 3D printing (MJP) is in the race with FDM. In comparison to FDM, MJP can work on a wide range of polymers and compound structures too.”

[2]. Chand, R.; Sharma, V.S.; Trehan, R.; Gupta, M.K. A physical investigation of dimensional and mechanical characteristics of 3D printed nut and bolt for industrial applications. Rapid Prototyp. J. 2022, 28(5), 953–966. DOI: https://doi.org/10.1108/RPJ-09-2021-0250

  1. Where are the PLA bolts used?

Examples with reference list:

- medical: bone screws (PLA bone screws containing iron oxide (Fe3O4) nanoparticles) [A1]

- medical: PLA in fixation plates and screws [A2]

- medical: Pins, suture anchors, and screws composed of PLA and copolymers [A3]

- toys, toy elements/manufacturing [9]

[A1] Narayanan G., Vernekar V.N., Kuyinu E.L., Laurencin C.T. Poly (lactic acid)-based
biomaterials for orthopaedic regenerative engineering, Adv. Drug Deliv. Rev. 2016, 107, 247–276.

[A2] DeStefano V., Khan S., Tabada A., Applications of PLA in modern medicine, Engineered Regeneration 1 (2020) 76–87

[A3] Wang, H.T.; Chiang, P.C.; Tzeng, J.J.; Wu, T.L.; Pan, Y.H.; Chang, W.J.; Huang, H.M. In vitro biocompatibility, radiopacity, and physical property tests of nano-Fe3O4 incorporated poly-L-lactide bone screws. Polymers 2017, 9, 191

[9] Eraliev O., Lee K-H.,  Lee Ch-H. Self-Loosening of a 3D-Printed Bolt by Using Three Different Materials under Cyclical Temperature Changes  Appl. Sci. 2022, 12(6), 3001; https://doi.org/10.3390/app12063001

  1. What is the basis for choosing strain amplitude and frequency. Please explain the scientific method behind choosing the parameters.

The classic method of loads carried by loosely fitted* (metric) bolts is the axial force (causing compression or stretching) with a static or variable value (in a small frequency range).

Typically, in the classical case where the threaded connection is made of metal alloys, the effect of deformation is rarely analyzed in the case of non-destructive testing. This is due to the fact that the said deformation is relatively small.

However, in the case of “plastics”, it makes sense as they can experience much greater deformation than metal materials without plastic deformation.

Thus, the analysis of the behavior of such a joint loaded with an axial force as a function of strain amplitude and frequency reflects its actual operating conditions.

What's more - PLA as a material has variable properties, which are also a function of the strain amplitude (in the case of pure PLA and its composites) and the frequency of loading (in the case of some PLA-based composites).

Since varying strain amplitude and frequency is the result of the work of the analyzed connection and at the same time has/may affect the material from which this connection is made, one can state that it is reasonable to analyze these parameters.

*Tight fitting screws are considered here to work in shear conditions (so basically pins with thread).

  1. Figure 2. Why only samples 1,4,2 are reported. Why not others? What are samples 1,2,4?

These are sample selected results. Figure 2 is and is intended to represent the rest of the results (show their variability and actual course). There is nothing special about samples 1, 2, 4 - it was decided to show only selected results for the sake of clarity. A graph with 6 curves or their multiples (12, 18, etc.) would not be legible at all.

  1. What does Stress values σ(3%) signify?

The value of compression force F(3%) and calculated on its basis compressive stress σ(3%)  are values corresponding to strain value of ε(3%) = 0.03. Value of about ε(3%) = 0.03determines the proportionality limit of stress and strain (see Fig. 2). The linear relationship between stress and strain is described by Hooke's law, an empirical model valid for some materials over a certain load and deformation. It can be said that the essence of the value ε(3%)  and σ(3%)  is the limit for which Hooke's law is valid for the recorded results. This value is significant from an engineering point of view and can be used during design.

  1. Figure 5. Calculated values of the force applied to one active thread ridge that cooperates with the nut F_n. Use subscripts wherever required. 

Editing error - has been corrected.

  1. Storage modulus of about ?′ = 3221 MPa. Please give reference papers wherein such high values are reported. I really doubt there is an error in the values reported. 

This value was read for the graph. It was double checked and there were no errors on our side. One’s guess is that in this particular article, there must be some error on the graph's axis. Nevertheless, this value is indeed quite high - to avoid repeating the error, this value has been removed from the manuscript.

  1. Conclusions are exhaustive. The need to be crisp and precise. Please rewrite the conclusions in point form.

Both Reviewers pointed out the Conclusions chapter for improvement (especially in the context of its length). For this reason, as suggested: it has been shortened, presented in bullet form and some numerical values have been removed from it. Additional emphasis was put towards the importance of the research and their applicability. The following text has been added to the text of the manuscript in place of the old one:

“The conducted research and analyzes allow us to conclude that:

 During the performed static compression tests, approximately 11% of all tested samples were damaged. The fracture failure (1 case) and ductile failure (all other cases) of the samples was observed. Thus, even in uniform conditions of production of threaded elements using the FDM method, it is difficult to explicitly predict the nature of failure or the value of the breaking stress. It is advisable to consider the value of safety coefficient when implementing bolt connections made of PLA.

 - The mean compression modulus value Ec   for all measurements was 917.79±184.99 MPa. The large value of standard deviation further supports the conclusion of a difficulty of precisely predicting the strength parameters of a printed thread element. 

 - The defined values of the compression modulus Ec differ from those specified in the literature by 2—2.5 times. The differences mentioned above were probably caused by the implementation of a threaded rod infill ratio of 35%.

 - The values of the storage modulus defined in cyclic tests increase E', while the values of the loss modulus E'' decrease when the value of the strain frequency increases.

 - Simultaneously the nature of the variations in storage modulus E' values in the function of the strain amplitude was not possible on the basis of the determined values. This issue remains open and requires further testing.

 The importance of the results obtained is emphasized by the fact that they complement the knowledge in the field of axial loads of elements made by 3D printing from PLA. This concerns particularly threaded connections, which can constitute a promising prospect in certain applications. The determined mechanical values can be used for designing threaded connections made of polylactic acid using the fused deposition modeling method.”

  1. A thorough English language and grammar checking is required for the entire manuscript.

The content of the revised manuscript was submitted for proofreading and corrected.

  1. Use of subscript and superscript is to be corrected throughout the manuscript and in the figures. 

Editing error - has been corrected.

  1. Figure 9. The value of (a) storage modulus E^' and (b) loss modulus E^'' as a function of strain amplitude ε. Please check the title of the fig.9.

That was indeed an error - thank You for noticing that. Of course it was supposed to be: "strain amplitude" - corrected.

  1. The trend in fig. 9 is erratic. What is the reason for this? 

It is actually a good question. This could be caused by different things.

First, it is theoretically possible that such results could be due to the fact that the sample was not stabilized. The first load cycles would cause work (shifting/slipping) and interaction of print layers. This probability has been eliminated by: a) introducing a preload, which eliminates the non-linearity of the results of the initial part of the force value recording; b) rejection of the peer measurements from the cycle due to the stabilization of the sample.

Secondly, such a phenomenon could be an effect of internal structure/bonds of material. This can be observed for some of PLA composites, for example for PLA/Cellulose nanocrystals composites [22], PLA/poly(caprolactone) blends, PLA/ Polycaprolactone, PLA/ poly (butylene adipate-co-terephthalate) and PLA/ poly-(butylene succinate-co-butylene adipate), etc. In such cases one can observe the dependency of dynamic modules on the strain amplitude in the case of increasingly small values - depending on the composition. Again, this is rather not the case, as in the discussed research PLA without any additives was analyzed.

The observed changes were perhaps caused by the fact that the printed structure was not solid and represented a triangle pattern infill structure and was therefore susceptible to strain. It is possible that the walls of the infill pattern experienced internal displacements that are not possible with a material with a solid structure - and that influenced the results.

Perhaps it is also the effect of a very small range values of the applied strain - which does not allow to draw general conclusions. For example, it is possible that the storage modulus values oscillate around some value that can be averaged, but this is not visible for the range 0.00-0.05 mm/mm (but it would be obvious e.g. for 0.00-10 mm/mm range).

As it was stated in paper - one cannot rule out that the general nature of the observed changes is still unknown due to the low values of the implemented strain amplitude. This issue remains open and requires further testing.

Reviewer 2 Report

The article is about compression strength of a PLA made bolts produced by the FDM method, However, some issues must to be addressed:

  1. Abstract: Please start by expressing the aim of this paper, followed by the rest of the information. Also, please define or try to avoid using abbreviations and long/useless numbers in the abstract. Typically, the abstract should provide a broad overview of the entire project, summarize the results, and present the implications of the research or what it adds to its field.
  2. The bibliographic foundation is important and well executed, however some new discussions should be inserted, authors should consider some works in the literature, such as: DOI 10.37358/mp.17.1.4793.
  3. Lines 145-174: The authors should rewrite the entire paragraph in order to exclude basis relations very well known in the scientific literature.
  4. The results are merely presented, not properly discussed. Please add explanations for the observed changes. Please give an extended discussion on the obtained results and correlate your findings with previous literature studies and prospective applications.
  5. More analysis and interpretation of the results should be added for a clearer understanding of observed experimental phenomena.
  6. The authors must to provide some details about importance of the research and their applicability.
  7. Please rewrite the conclusions in a more quantitative form and enhance the clarity of the conclusion section (too many numbers) in order to highlight the results obtained.
  8. General check-up and correction of the English language is suggested. There are still some minor typos and grammatical errors.

The author needs to address the abovementioned points for the betterment of the manuscript.

Author Response

The article is about compression strength of a PLA made bolts produced by the FDM method, However, some issues must to be addressed:

  1. Abstract: Please start by expressing the aim of this paper, followed by the rest of the information. Also, please define or try to avoid using abbreviations and long/useless numbers in the abstract. Typically, the abstract should provide a broad overview of the entire project, summarize the results, and present the implications of the research or what it adds to its field.

The abstract has exactly the structure requested by the reviewer - that's why no changes were made to the formula. It has a classic structure: Purpose (the aim of this paper is stated clearly in the first sentence of Abstract: “The aim of this research was to define...” ), Methods, Results, Implications (The last sentence of Abstract: (“The determined mechanical values can be therefore used for…”) Abbreviations were removed from the text, as well as some results in numerical form – as suggested (only the average compression modulus value was left as the most important value).

  1. The bibliographic foundation is important and well executed, however some new discussions should be inserted, authors should consider some works in the literature, such as: DOI 10.37358/mp.17.1.4793.

The indicated literature was reviewed and added to the bibliography list.

  1. Lines 145-174: The authors should rewrite the entire paragraph in order to exclude basis relations very well known in the scientific literature.

Ones guess is, this remark concerns stress and strain formulas. Suggested changes have been made. Equation 1 was left unchanged as it may not be obvious to the reader.

  1. The results are merely presented, not properly discussed. Please add explanations for the observed changes. Please give an extended discussion on the obtained results and correlate your findings with previous literature studies and prospective applications.
  2. More analysis and interpretation of the results should be added for a clearer understanding of observed experimental phenomena.
    (both points)

(line numbering and bibliography for clarity according to uncorrected manuscript)

Results of static tests – compression modulus Ec

The determined values are presented in the form of charts and tables. Then, the text compares the determined value of compression modulus Ec with the values found in the literature. Cases considered to be the most similar in the context of research methodology and sample preparation were analyzed (bibliography [15-19] - before changes). Explanation of the differences and explanations for the observed changes are included in lines 250-263

Results of static tests – values of forces and stress in the context of the thread and failure of samples

The determined values were presented in the form of graphs and photos of damaged samples. Analysis and interpretation of the results concerning the first issue is presented in 267-273. An attempt to explain the failure mechanism is contained in lines 287-292. A comparison with the literature reference [5] in terms of the most probable cause of sample failure is also included.

Results of cyclic tests – dynamic moduli E’ and E”

As far as correlation of findings with previous literature studies is considered – the matter is a bit complicated. In the case of the results of cyclical studies, a direct comparison of the results with the results of the work of other researchers is not possible. It was not possible to find results describing identical/very similar studies in this aspect (which may indicate their innovative nature). What is stated and explained in lines 346-348:

“The authors were not able to find any results of other experiments that would be comparable with those presented in this article. A direct comparison was therefore not possible, nevertheless an attempt was made to refer to literature values.”

Of course, reference to the current state of knowledge is very important, which is why it was decided to select the most similar results in the context of the material and the determined values. As a result, papers [25-28] were analyzed and the results were discussed. In this case, the main difficulty was that the reported values generally referred to the change of dynamic modulus during loading in the tangential plane (G' and G") and solid structure materials (as opposed to the incompletely filled samples used). The selection of available literature is limited because very often incomplete parameter values are given (e.g. characteristic G' but not G'' and so on).

An attempt to explain the observed experimental phenomena was included after comparing the results and in individual aspects was included in the following lines: 340-345 (changes of E' and E" with strain amplitude), 376-380 (difference in the scale of results), 311-315 (changes od E ' and E" with frequency)

Only one of the three reviewers made this kind of remark, which is somewhat surprising. Anyway, the content indicated by the reviewers was added to the manuscript (along with new research on the tested material). The Authors hope that the revised version meets expectations and given requirements.

  1. Please rewrite the conclusions in a more quantitative form and enhance the clarity of the conclusion section (too many numbers) in order to highlight the results obtained.

Both Reviewers pointed out the Conclusions chapter for improvement (especially in the context of its length). For this reason, as suggested: it has been shortened, presented in bullet form and some numerical values have been removed from it. Additional emphasis was put towards the importance of the research and their applicability. The following text has been added to the text of the manuscript in place of the old one:

“The conducted research and analyzes allow us to conclude that:

 During the performed static compression tests, approximately 11% of all tested samples were damaged. The fracture failure (1 case) and ductile failure (all other cases) of the samples was observed. Thus, even in uniform conditions of production of threaded elements using the FDM method, it is difficult to explicitly predict the nature of failure or the value of the breaking stress. It is advisable to consider the value of safety coefficient when implementing bolt connections made of PLA.

 - The mean compression modulus value Ec   for all measurements was 917.79±184.99 MPa. The large value of standard deviation further supports the conclusion of a difficulty of precisely predicting the strength parameters of a printe

- The defined values of the compression modulus Ec differ from those specified in the literature by 2—2.5 times. The differences mentioned above were probably caused by the implementation of a threaded rod infill ratio of 35%.

 - The values of the storage modulus defined in cyclic tests increase E', while the values of the loss modulus E'' decrease when the value of the strain frequency increases.

 - Simultaneously the nature of the variations in storage modulus E' values in the function of the strain amplitude was not possible on the basis of the determined values. This issue remains open and requires further testing.

 The importance of the results obtained is emphasized by the fact that they complement the knowledge in the field of axial loads of elements made by 3D printing from PLA. This concerns particularly threaded connections, which can constitute a promising prospect in certain applications. The determined mechanical values can be used for designing threaded connections made of polylactic acid using the fused deposition modelling method.”

  1. The authors must to provide some details about importance of the research and their applicability.

Part of the answer to this question is contained in the above point under the Conclusions section (above). Paragraphs related to the importance of the research and their applicability have also been added and incorporated into the manuscript. The following text has been added:

“Current applications of PLA made bolts and its composites include medical aspects: PLA bone screws containing iron oxide (Fe3O4) nanoparticles [A1], PLA in fixation plates and screws [A2], pins, suture anchors, and screws composed of PLA and copolymers [A3]. One can also indicate applications as toys and toy elements [9]. It is possible that further study of this type of connections will contribute to their wider use in future.

The choice of PLA can be justified because in the case of FDM it is one of the most commonly used materials. It has many advantages, including biodegradability, biocompatibility and environmentally friendly properties [D1, D2]. At the same time, the use of pure PLA also has disadvantages such as relatively low mechanical properties, water solubility rate or high brittleness as well as low heat resistance [D1, D2].”

  1. General check-up and correction of the English language is suggested. There are still some minor typos and grammatical errors.

The content of the revised manuscript was submitted for proofreading and corrected.

The author needs to address the abovementioned points for the betterment of the manuscript.

We appreciate the time and effort and are grateful for the insightful comments and valuable improvements to our work. We carefully considered the comments and tried our best to address all of them. In addition, many other corrections have been made. All changes are marked in “change tracking

Reviewer 3 Report

Dear Authors,

I studied your manuscript entitled "Compression strength of a PLA made bolts produced by the FDM method". Although the approach is interesting, some spaces need to be improved in terms of journal quality. I recommend minor revision before further consideration for publication in the Materials.

1) The research question is not very well stated and discussed.

2) More details about the materials should be provided. The content of D-LA in PLA should be provided if the author can get this information.

3) The results of WAXD analysis could be reported and discussed to determine the crystallographic properties of the samples.

4) PLA is subjected to a facile thermal and mechanical degradation during sample preparation. Can the authors assume (with experimental support) that their results are free of these problems?

5) The text should be reviewed thoroughly to remove some incorrect or confusing phrases.

Author Response

I studied your manuscript entitled "Compression strength of a PLA made bolts produced by the FDM method". Although the approach is interesting, some spaces need to be improved in terms of journal quality. I recommend minor revision before further consideration for publication in the Materials.

We appreciate the time and effort and are grateful for the insightful comments and valuable improvements to our work. We carefully considered the comments and tried our best to address all of them. In addition, many other corrections have been made. All changes are marked in “change tracking.

1) The research question is not very well stated and discussed.

The content related to the aim of the research was broadened and additionally discussed. The following text has been added to the manuscript:

“In view of the above, the aim of this research was to determine the compression strength of PLA made bolts produced using the FDM method. For this purpose, static and cyclic compression tests were carried out on metric threads of selected sizes. It was considered crucial to determine the values  of compression modulus , storage modulus  the loss modulus . Threaded connections usually experience axial force with a static or variable value resulting in varying strain amplitude and frequency of load. It is therefore reasonable to analyze material parameters in this aspect. Previous literature studies, in terms of axial stress, focus mainly on the tensile strength of PLA. For this reason, the analysis of compressive strength is justified – which complements the state of knowledge in this field.

2) More details about the materials should be provided. The content of D-LA in PLA should be provided if the author can get this information.

Based on the reviewer's comment, the manufacturer's data related to the properties of the tested material were reviewed. The following text has been added to the body of the manuscript:

Filament Fiberlogy Refill Easy PLA (Fiberlab S.A., Brzezie, Poland) was used to produce the test samples. It is commercially available material that can be considered typical. Mechanical properties of this material declared by the supplier are presented in Table 1.

Table 1. The properties of tested materials.

Property         Testing method                       Value

Specific Density                     ISO 1183        1.24 g/cm3

Tensile Strength (Yield)                     ISO 527                      50 MPa

Tensile Strength (Break)                    ISO 527                      53 MPa

Tensile Modulus         ISO 527                      3500 MPa

Elongation (Yield)     ISO 527                      6%

Flexural Strength       ISO 178                      81 MPa

Flexural Modulus       ISO 178                      3800 MPa

Izod Impact Strength (Notched, 23 °C)         ISO 180                      2 kJ/m2

It is true that The L/D-isomer ratio in PLA affects a number of mechanical, rheological and thermal properties of this material. It is understood that in research works concerning production methods of different variations of this material, the exact chemical composition and synthesis methods are given. This also applies to PLA composites that aim at modifying final product properties. This is justified for many reasons, but it is also intended to give the possibility of possible replication of research results.

Precise information on the composition of the material in the context of chemical preparation/synthesis is not available. This basically constitutes the know-how of production companies and as such cannot be disclosed.

The nature of this study is primarily application, in the context of the practical use of generally available methods and material. Possible replication is possible by obtaining commercially available material for printing, because it is publicly available.

3) The results of WAXD analysis could be reported and discussed to determine the crystallographic properties of the samples.

The comment of the Reviewer was analysed and, as a result, additional research was performed. Their results and descriptions have been added to the manuscript.

In order to determine the crystallographic properties of the samples, X-Ray Diffraction (XRD: X-Ray Diffraction) was carried out. They were conducted using a powder diffractometer (SmartLab Rigaku, Japan) with a CuK α lamp, in the angle range of 3-80° (2θ) in steps of 0.04°. A square print sample with a side length of 20 mm was selected for testing. This sample was made with the same printing parameters as the tested elements. However, it was made as a solid element (100% infill rate) for test accuracy and in order to achieve the greatest certainty of results as possible. The XRD measurements were performed at room temperature. The test results are shown in Fig. XXX. As can be seen in the diffraction pattern, the PLA filament used in 3D printing is not completely amorphous. Amorphousness is indicated by the presence of a broad diffraction peak at the maximum at the angle of 16.1° (2θ) and at the angle of 33.3° (2θ) [16]. Reflections present at angles: 27.2° ; 35.5°; 55.0° (2θ) may indicate the presence of colored PLA additives [17].

 Fig. 1 Diffractogram obtained for the tested PLA filament (see in the corrected manuscript).”

[16] Fuß, F.,  Rieckert, M.; Steinhauer, S.; Liesegangb, M.; Thielea, G. 3D-printed equipment to decouple (powder) X-ray diffraction sample preparation and measurement, J. Appl. Cryst. 2022, 55, 686–692. DOI: https://doi.org/10.1107/S160057672200293X.

[17] Pearce, J.; Wittbrodt, B. T. The effects of PLA color on material properties of 3-D printed components. Addit. Manuf. 2015, 8, pp.110–116. DOI: https://doi.org/10.1016/j.addma.2015.09.006.

4) PLA is subjected to a facile thermal and mechanical degradation during sample preparation. Can the authors assume (with experimental support) that their results are free of these problems?

Again, the research was intended to be practical/applicable. Test samples were made from generally available materials and a commercially available 3D printer. Thus, an objective measurement in the indicated aspect would be difficult to carry out. In terms of changing the properties of the material during printing. Since all samples were made in the same way (with the same printing parameters) - it can be stated that even if there was thermal and mechanical degradation, it applies to all samples to the same/similar extent. For this reason, it does not affect the general character of the obtained results. The fact is that 3D printing of this type of material will affect its properties. An analysis of this issue is beyond the scope of this article. Nevertheless, this is a valid point - it will be a good starting point for future research and analysis in this aspect.

5) The text should be reviewed thoroughly to remove some incorrect or confusing phrases.

The content of the revised manuscript was submitted for proofreading and corrected.

Round 2

Reviewer 1 Report

Authors have addressed the comments. I now recommend publication.

Reviewer 2 Report

The article is suitable for publication.